# NEOPRENE v1.0.1: A Python library for generating spatial rainfall based on the Neyman-Scott process

Javier Diez-Sierra[1,2,3], Salvador Navas[1], and Manuel del Jesus[1]

[1]IHCantabria - Instituto de Hidráulica Ambiental de la Universidad de Cantabria, Santander, Spain.
[2]Instituto de Física de Cantabria (IFCA), CSIC-Universidad de Cantabria, Santander, Spain.
[3]Dept. of Applied Mathematics and Computer Science (MACC), Universidad de Cantabria, Santander, Spain

**Correspondence:** Manuel del Jesus (manuel.deljesus@unican.es)

**Abstract.** Long time series of rainfall at different levels of aggregation (daily or hourly in most cases) constitute the basic input for hydrological, hydraulic and climate studies. However, often times the length, completeness, time resolution or spatial coverage of the available records fall short of the minimum requirements to build robust estimations. Here, we introduce NEO-PRENE, a Python library to generate synthetic time series of rainfall. NEOPRENE simulates multi-site synthetic rainfall that reproduces observed statistics at different time aggregations. Three case studies exemplify the use of the library, focusing on extreme rainfall, as well as on dissaggregating daily rainfall observations into hourly rainfall records. NEOPRENE is distributed from GitHub with an open license (GPLv3), free for research and commercial purposes alike. We also provide Jupyter notebooks with the example uses cases to promote its adoption by researchers and practitioners involved in vulnerability, impact and adaptation studies.

## 1 Introduction

Stochastic rainfall models are used in hydrological, hydraulic and climate studies because rainfall records at ground stations are often inadequate for applications in terms of their length, completeness, time resolution or spatial coverage. These models are able to generate arbitrarily long time series of synthetic rainfall that reproduce different observed rainfall statistics (i.e., means, variances and covariances, frequencies, extremes, spatial and temporal correlation, etc.) at different levels of aggregation (Cowpertwait, 2006; Cowpertwait et al., 2013). Stochastic rainfall models and weather generators have also been used in water resources assessments (Alodah and Seidou, 2019; Kiem et al., 2021; Fowler et al., 2000), landslide analysis (Thomas et al., 2018) and urban flooding assessment (Park et al., 2021).

A number of stochastic rainfall models have been developed in the last decades based on different statistical techniques such as Poisson-gamma models, Markov models, Monte-Carlo models and Bayesian Networks models, among others (Legasa and Gutiérrez, 2020; Kleiber et al., 2012; Wilks and Wilby, 1999). Poisson clustered models are among the most commonly used due to their flexibility and the high degree of accuracy they provide (Rodriguez-Iturbe and Eagleson, 1987; Cowpertwait et al., 2013; Burton et al., 2010; Fowler et al., 2005; Leonard et al., 2008). These models are based on a Poisson process of storm origins which have a random number of rectangular pulses ("rain cells") associated with them, with heights corresponding to

rain intensity and widths to cell duration. Different cells and storms may overlap so that the total rain intensity at any time is the sum of the intensities of all cells active at that time (Cowpertwait et al., 2002).

Considerable research on the modeling of rainfall has been undertaken using two different Poisson clustered approaches: Neyman-Scott and Bartlett-Lewis. Several studies have demonstrated that both approaches, which differ in the displacement of cell origins relative to storm origins, are able to reproduce observed rainfall statistics, including second order properties (see Islam et al., 1990; Cowpertwait, 1991, 1995; Cowpertwait et al., 1996b, a; Cowpertwait, 1998; Kaczmarska et al., 2014; Onof and Wang, 2020; Park et al., 2019; Kim and Onof, 2020).

Poisson clustered models are useful for many purposes, specially in Engineering practice (e.g. return period estimation for flood analysis) and have been used for applications in hydrology (Puente et al., 1993) and tested to evaluate their suitability to represent extreme events (Verhoest et al., 2010). However, their main use so far has been related to analyzing rainfall itself (Onof and Wheater, 1994; Cowpertwait and O'Connell, 1997; Diez-Sierra and del Jesus, 2019) probably due to the difficulty of properly implementing these models *ex novo*.

Indeed, there is a lack of readily available software solutions implementing this kind of models, which severely limits its usefulness to the general scientific and technical community. To our knowledge, only three software tools have been developed to date: RainSim (Burton et al., 2008), HyetosR (Kossieris et al., 2012), and Let-It-Rain, which has a web version (Kim et al., 2017) as well as a desktop version (Kim and Onof, 2020). RainSim is based on the Neyman-Scott process and HyetosR and Let-It-Rain on the Bartlett-Lewis one.

RainSim is able to simulate multi-site stochastic rainfall at different temporal aggregations and uses the Shuffled Complex Evolution (SCE-UA, Duan et al. (1992)) optimization algorithm for calibration. Its major limitation, in our opinion, is its availability (under demand from the authors, only for research purposes and for one specific operative system). HyetosR, implemented in R, may simulate stochastic time series of rainfall that reproduce several observed statistics at daily and hourly temporal aggregation (mean, variance, covariance and probability of a dry period). In contrast to RainSim, it is easily available since it is distributed with an open-source library. However, HyetosR only deals with point precipitation, so it cannot be used to generate rainfall fields.

The web version of Let-It-Rain can be used to simulate synthetic point rainfall time series at hourly resolution for the United States and Korea, using a modified Bartlett-Lewis rectangular pulse model. It also presents a regionalization that allows the user to generate synthetic time series even at ungauged locations. The desktop version is a later development that allows the user to reproduce rainfall characteristics from very short time scales (5 minutes) to very long ones (decades). The software is provided as a compiled executable for the MS Windows operative system.

The NEOPRENE library constitutes —to the best of our knowledge— the first open-source library for stochastic rainfall generation based on the spatiotemporal Neyman-Scott process. The open-source GPLv3 ensures that the code can be freely use for research and commercial purposes. NEOPRENE is readily available for all major operative systems from its GitHub repository (Diez-Sierra et al., 2021a) and from Zenodo (Diez-Sierra et al., 2021b), that enables long-term archival of repository snapshots. NEOPRENE simulates synthetic multi-site rainfall at different temporal aggregations that reproduce different observed rainfall statistics. NEOPRENE can be used for multiple purposes such as extreme rainfall analysis or rainfall disag-

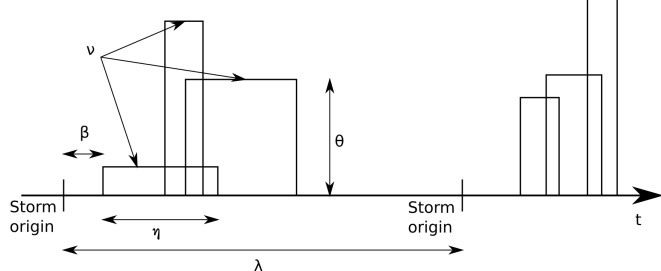

**Figure 1.** Model scheme showing the meaning of the main parameters.

gregation, among others. NEOPRENE is designed to reproduce second order moments (see Cowpertwait, 1998) and allows to
simulate several storm systems simultaneously —each system with its own set of parameters (see (Leonard et al., 2008))— to
capture different rainfall generation processes (i.e. frontal and convective precipitation), making NEOPRENE a powerful tool
for rainfall extreme analysis and for vulnerability, impact and adaptation (VIA) studies.

## 2  Methods

### 2.1  Mathematical model description

Point processes based on clustered Poisson models for rainfall modeling have been widely presented in scientific literature. A
good introduction to Neyman-Scott models can be found in Cox and Isham (1988). A detailed description of the mathematical
model behind the NEOPRENE library can be consulted in Cowpertwait et al. (2013), although additional details can be found
in Cowpertwait (1995) and Leonard et al. (2008). del Jesus et al. (2015) also presents some interesting derivations; mainly
related to parameter fitting from satellite observations and the characterization of dry periods. Diez-Sierra and del Jesus (2019)
present a methodology for subdaily rainfall estimation through daily rainfall downscaling using Neyman-Scott models.

In this subsection, a brief description of the mathematical model is provided; enough to understand which are the model
parameters, their effects and the results generated by the library. Readers interested in a more exhaustive inspection of the
mathematical innards of the library are invited to consult the references provided above and the documentation section available
in the GitHub repo (Diez-Sierra et al., 2021a).

The model used for the NEOPRENE library assumes that rainfall at a given region occurs by superposition of different types
of storms ($S_i$); each storm represented by an independent point process. The NEOPRENE library allows a maximum of two
storm types (or independent point processes), which tends to be sufficient for most applications. For instance, in many places
this decomposition may serve to capture frontal ($S_1$) and convective ($S_2$) precipitation. Each type of superposed process ($S_i$)
would represent some proportion ($\alpha_i$) of the total number of storms.

The interarrival time between the origins of storms of type $i$ follows an exponential distribution with paramater $\lambda_i$. Each
storm is composed of several rainfall cells, which form a marked point process. Each rainfall cell is assumed to be circular and

to contain a random amount of water, that produces rainfall at a random rate (cell intensity) during the lifetime of the rainfall cell (cell duration). The superposition of all these cells and their combined intensities produces the total rainfall intensity of the model. The marked point process generated by the rainfall cells is characterized by:

- $U_i$ and $V_i$, the 2D coordinates of the cell center, that follow a 2D Poisson process with rate $\Lambda_{ci}$ (per unit area).

- $R_{ci}$, the radius of the cell, that follows an exponential distribution with parameter $\rho_{ci}$.

- $L_i$, the lag from the storm origin to the origin of the cell, that follows an exponential distribution with parameter $\beta_i$.

- $D_i$, the duration of the rainfall cell, the time during which the storm produces rainfall, that follows an exponential distribution with parameter $\eta_i$.

- $I_i$, the rainfall intensity of the cell, that may follow different distributions, but that is usually taken to follow an exponential distribution of average $\theta_i$ (or parameter $1/\theta_i$).

Rainfall occurs at any give location and time if, and only if, a rain cell covers that point during that time. The total rainfall intensity at any given time and location is the sum of the rainfall intensities induced by all the rain cells active at that point at that time. The total number of rain cells covering a point follows a Poisson distribution with parameter $\nu_i = 2\,\pi\,\Lambda_{ci}/\rho_{ci}^2$.

As some relations exist among all the above mentioned parameters, a set of 6 parameters: $\lambda_i, \nu_i, \beta_i, \eta_i, \rho_{ci}, \theta_i$, is sufficient to represent any storm type. When using two different storm types, all the characteristics of the rainfall process are captured by a set of 12 parameters, 6 for each storm type.

Note that in the case of the spatio-temporal model, we work with normalized statistics, making necessary to introduce an additional parameter, $\xi_i$, which is estimated for each period using the observed average rainfall. $\xi_i$ acts as a scale parameter, that captures the differences in average precipitation at different locations (gage location). $\xi_i$ serves to reproduce the gradients of average rainfall in the area of interest. This parameters is adjusted on a site-by-site basis once the rest of the parameters have been estimated (Cowpertwait et al., 2013).

## 2.2   Aggregated properties

The Neyman-Scott model represents a continuous process in space and time. However, any rainfall measurement —rain gage or satellite observations— aggregates information; in time —the former— and in time and space —the latter—. Therefore, the properties of the aggregated process are necessary to compare the model and the observations. There is also need to derive some aggregated properties because the model is a simplified conceptualization of the rainfall process, and some of the properties defined —e.g. the cell radius, or the cell intensity— cannot be measured independently; only their effects can be measured.

The integrated properties of the model are presented in the references provided at the beginning of the section(Cox and Isham, 1988; Cowpertwait et al., 2013; Cowpertwait, 1995; del Jesus et al., 2015). The properties of the aggregated process can be derived using the theory of random fields (Vanmarcke, 2010). Integration over space and time would result in a process similar to satellite rainfall observations, while integration over time only would result in the equivalent of rain gage observations.

The library allows to fit the aggregated average (mean), variance, temporal autocorrelation, probability of no rainfall, transition probabilities between two successive wet periods or two successive dry periods, skewness and cross-correlation. The formulas for these aggregated statistics, as well as their derivation process, were obtained from the following references (Cowpertwait, 2006; Cowpertwait et al., 2013; del Jesus et al., 2015) and they are available in the documentation section available on the GitHub repo (Diez-Sierra et al., 2021a).

## 2.3 Parameters fitting and rainfall simulation

NEOPRENE fits the model parameters by minimizing the weighted Euclidean distance between the observed statistics and the modeled ones. Any subset of all the possible aggregated statistics (multiple of a daily or an hourly temporal aggregation) may be used for fitting. It is important to remind here that observations do not belong to the continuous point process but to the aggregated one, so the observed statistics cannot be directly equated to the point process statistics.

The weights for the weighted Euclidean distance can be freely chosen by the user. Particle Swarm Optimization (PSO, Kennedy (2011)) is used for fitting (which is a minimization process). The result of the fitting procedure is the set of optimal parameters.

Once the model parameters are defined, time series generation is a straightforward process. Storm arrivals are simulated following a Poisson process with parameter $\lambda_i$. The number of cells corresponding to the storm are simulated also with a Poisson process, this time with parameter $\nu_i$. Then, for each cell, four values are obtained from four exponential distributions with parameters $\beta_i$, $\eta_i$, $\rho_i$ and $\theta_i$, corresponding to the cell lag (with respect to the storm origin), its duration, its radius and its average intensity.

Repeating this process in time, a time series of total precipitation intensity can be generated for all the points in a given domain, or for any given isolated point.

Seasonality is included in the model accounting for different sets of model parameters and observed statistics. For instance, a single set of parameters will be used to calibrate and generate from a model without seasonality, where the year is assumed to be a stationary period. However, in regions where two well differentiated periods may be observed -a wet and a dry season, for instance-, two different sets of parameters will be used: one for the dry season and another one for the wet season. That is, seasonality is accounted for by decomposing the complete time series into subseries that only contain the information related to the desired season or time period.

## 3 The NEOPRENE library

The NEOPRENE library implements the Neyman-Scott process for the analysis of spatio-temporal rainfall, that is, spatial fields of rainfall can be captured or simulated as they change over time. The model can also be used to reproduce rainfall at a specific point, without taking in consideration the behavior of rainfall in the surroundings.

Rainfall generation can be decomposed in two steps: a calibration step and a simulation step (as shown in Figure 2). The calibration step serves to find the set of parameters that best reproduces the statistical properties of the series given as an input,

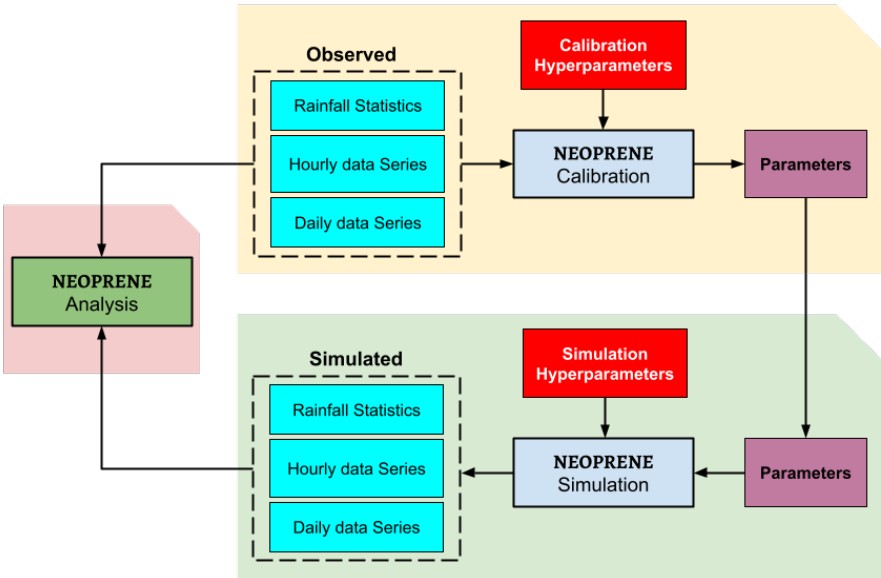

**Figure 2.** Schematic representation of the three main modules implemented in the NEOPRENE library: calibration, simulation and analysis. Observed time series (daily or hourly) or observed statistics need to be provided by the user to the calibration module, which returns the optimal set of parameters. The simulation module uses the optimal parameters to generate arbitrarily long time series of synthetic rainfall data that reproduce different observed rainfall statistics. Daily and hourly time series, as well as their statistics, are returned in all cases by the simulation module. Observed and simulated rainfall time series can be compared with the analysis module which also contains some functions for daily-to-hourly rainfall disaggregation. Calibration and simulation hyperparameters are required to define, for instance, the maximum number of calibration iterations, the statistics and aggregation levels that have to be fitted and simulated, or the starting and ending dates for a simulation.

or that best match the provided rainfall statistics. The simulation step takes a set of parameters as input and reproduces a time series of the process (punctual or spatial) that follows the supplied parameters. Additionally, the NEOPRENE library provides several functions for validation and for daily-to-hourly rainfall disaggregation.

The `disaggregate_rainfall` function (see section 3.1.3) performs the disaggregation process. This function scans the observed daily time series. For each day, the function looks in the synthetic time series for the most similar day to the one being dissagregated —the one being selected in the observed time series—. To improve the quality of the dissagregation, previous days are also used in the search. The series of days that minimize the Euclidean distance between the observations and the aggregated synthetic time series are selected, and the hours that constitute that day are used for the dissagregated time series. The process is then repeated for each day until the complete observed time series has been dissagregated. This function is also implemented for the multi-site model. A more detailed explanation of the dissagregation process can be found at Cowpertwait (2006).

The normal use case for the library would be to configure the calibration process —setting the hyperparameters of the calibration process— and then provide the observed data to the calibrator. The calibrator would look for the optimal set of parameters, where the definition of *optimum* can be tweaked through the hyperparameters. Once finished, the calibrator would provide a set of optimal parameters.

Hyperparameters are all those parameters that do not belong to the Neyman-Scott process but that are required to configure either the calibration or the simulation steps. Such parameters may be the maximum number of calibrations iterations or the starting and ending dates for a simulation.

Then, a simulation should be configured, again through the use of hyperparameters. The simulation step requires also to define the time coverage of the simulated time series. The simulator receives, in addition to the hyperparameters, the set of optimal parameters, and use them to generate a time series of rainfall.

In some cases, the same set of optimal parameters may be used to generate different time series, either with varying hyperparameters, either with small modifications of the optimal parameters, for instance for a sensibility analysis. The library is flexible enough to adapt to many possible use cases.

It is important to note that the underlying mathematical model is, in our own experience, flexible enough to adapt to different combination of the rainfall statistics. Therefore, it should be able to properly model rainfall for different climates. The main limitation being in locations where more than two types of precipitation occur. In this cases, the model may struggle to provide an optimal performance.

## 3.1 Library implementation

The library has been implemented following the two step operation described so far: there is a "calibration" sublibrary and a "simulation" one. In general, both steps will be followed sequentially, but in some cases (the evaluation of several life cycles of a given infrastructure, for instance) several simulation steps may be carried out connected to only one calibration step. A third sublibrary, "analysis", contains several functions to extend the functionality of the library and to simplify its use, like a

function to compare the simulated series with the observed ones and a function for daily-to-hourly rainfall disaggregation, for instance.

The library contains two main python classes, `NRSP` and `STNRSP`, which allow to simulate single-site and multi-site synthetic rainfall series, respectively. The first python class calculates the observed rainfall statistics from a single time series and simulates an arbitrarily long time series of synthetic rainfall which reproduces the observed statistics. This class is able to reproduce the following statistics for any daily or hourly temporal aggregation: mean, variance, temporal autocorrelation, probability of no rainfall, transition probabilities between two successive wet periods or two successive dry periods and the skewness. The second python class calculates the average rainfall statistics from a list of series and simulates a synthetic list of rainfall series which reproduce observed statistics averaged over all the series (except for the mean statistic which varies in space to mimic rainfall intensity fields). Additional to the above statistics, the `STNRSP` is able to reproduce the cross-correlation.

Figure 2 shows a schematic representation of the three main classes implemented for the NEOPRENE library: calibration, simulation and analysis. These classes are described in depth in the following subsections.

### 3.1.1 Calibration

The calibration step is implemented within the calibration sublibrary , specifically in the `Calibration` python class (one within `NSRP`, for the point model, and another within `STNSRP` for the multi-site model). It requires as input a single time series or multiple ones, or the observed statistics. It outputs the calibrated optimal parameters needed for the simulation.

It is important to note that, internally, the library calibrates against the specified statistics. Therefore, in order to ensure a good calibration and representation of the areal rainfall, the length of the time series as well as its completeness should allow a robust computation of any of the selected statistics. In general terms, the amount of information required for calibration will depend on the final applications of the data. If rainfall extremes are desired, then at least 30 years of data should be collected. If missing data are below an acceptable threshold (20% of the overall length of the time series), no data filling should be required.

The calibration process is controlled by a set of parameters —that we will call hyperparameters— that should be set by the `Calibration` python class of the hyperparameters sublibrary (`HiperParams`, again one within `NSRP` and another within `STNSRP`). The following calibration hyperparameters can be set:

- **Data:** A pandas dataframe containing the original time series.

- **Seasonality:** Python list that configures the desired seasonality.

- **Temporal resolution:** String that defines the temporal resolution of the provided time series (hourly and daily temporal resolution can be provided).

- **Process:** String configuring whether one or two storm systems should be considered.

- **Statistics:** Python list of string that contain the statistics that have to be considered during the fitting process.

- **Weights:** List that contains the weight for computing the total error —Euclidean distance— between the observed statistics and the generated ones.

- **Number of iterations:** Integer that defines the maximum number of iterations of the calibration process.

- **Number of bees:** Integer that defines the number of particles to use in the PSO algorithm.

- **Number of initializations:** Integer that defines the number of initializations to be performed during the calibration procedure.

- **Time between storms:** Range of acceptable values of storm interarrival times (in hours).

- **Storm cell displacement:** Range of acceptable values of cell lags (in hours).

- **Number of storm cells:** Range of acceptable values for the number of cells per storm.

- **Cell duration:** Range of acceptable values for the duration of a storm cell (in hours).

- **Cell intensity:** Range of acceptable values for the intensity of a storm cell (in mm/hour).

- **Coordinates:** String defining the type of coordinates: geographical (in degrees) or UTM (in meters).

- **Cell_radius:** Range of acceptable values of the cell radius (in km).

The hyperparameters "coordinates" and "cell_radius" are only required for the multi-site model (STNSRP).

The **Number of iterations** and **Number of bees** hyperparameters control how exhaustive the search is in the parameter space for the optimal solution. In our experience 100 iterations and 1,000 bees are a good minimal value set to get good results. Additional advice may be found in specific literature (Kennedy, 2011).

The **Number of initializations** hyperparameters allow the library to restart the search multiple times to ensure that the search did not get trapped in a local minimum. This is almost never the case, but the number of initializations may be increased in cases where the initial results seem suboptimal. Indeed, we recommend increasing this hyperparameter before increasing the number of bees or iterations.

The hyperparameters that refer to physical properties of the storm itself (**Time between storms**, **Storm cell displacement**, **Number of storm cells**, **Cell duration**, **Cell intensity** and **Cell radius**) should be used to ensure that reasonable values are obtained. To set these parameters, a minimum knowledge of the properties of the rainfall process in the specific location being analyzed is required. The **time between storms** normally represents the time lag that separates independent storms, while **storm cell displacement** captures the time lag between rain cells belonging to the same storm. Similarly, **Cell duration** captures the time lag during which rainfall intensity is constant, **Cell intensity** captures the range of possible intensities at a site and **cell radius** represents the maximum length that may be affected by a given storm. The reader is advised to consult some of the included references (Isham et al., 2005, for instance) to obtain a deeper grasp on the selection of these hyperparameters.

### 3.1.2 Simulation

The simulation step is implemented within the simulation sublibrary (`Simulation`, one for the point model and another for the multi-site one), specifically in the `Simulation` python class. It requires as input the calibrated parameters and return the simulated rainfall series at both daily and hourly temporal aggregations.

The simulation process is controlled by a set of hyperparameters that should be set by the `Simulation` python class of the hyperparameters sublibrary (`HiperParams`, again one for the point model and another for the multi-site one). The following simulation hyperparameters can be set:

- **Parameters_simulation:** The values of the parameters, usually the calibrated ones.

- **Year_ini** and **year_fin:** The initial and final years of the simulation.

- **Seasonality, temporal resolution, process, statistics:** The simulation and the calibration sublibraries are fully independent, thus several hyperparameters are necessarily repeated.

### 3.1.3 Analysis

The analysis module is not required for either rainfall calibration or simulation, but it is helpful for many tasks such as to check the performance of the model or for rainfall disaggregation, for instance. This functionality is implemented within the analysis sublibrary (`Analysis`, one for the point model and another one for the multi-site one), specifically in the `Analysis` python class.

To evaluate the quality of the fit, the user should determine which statistical test may be appropriate in every case. A Kolmogorov-Smirnov test could be suitable to test if the generated rainfall and the observed one come from the same distribution, while a t-test could serve to analyze if the difference in mean precipitation is significant. The number of possible hypothesis tests that may be carried out over the model results is huge, since different applications may require different quality evaluation. Therefore, we do not provide any goodness-of-fit routine or specific hypothesis testing within the Analysis module. However, we do provide functions to help to inspect the quality of the fits.

The `Analysis` python class contains several functions for validation and for daily-to-hourly rainfall disaggregation. So far the functions implemented are:

- **`compare_statistics_fig()`:** This function returns an image with the observed (Obs), fitted (Fitted) and simulated (Sim) statistics (see Figure 3).

- **`exceedance_probability_fig()`:** This function returns an image comparing the exceedance probability for the observed and simulated series (see Figure 4).

- **`correlation_fig()`:** This function returns an image with the observed (Obs), fitted (Fitted) and simulated (Sim) cross-correlation (see Figure 7).

- **disaggregate_rainfall and disaggregation_fig:** The first function is used for daily-to-hourly rainfall disaggregation (see Cowpertwait (2006)). The second function returns an image comparing the observed rainfall time series and the disaggregated ones (see Figure 5).

– **save_figures:** This function allows to save the figures in a folder.

## 4   Use cases

In this section three use cases for the library are introduced. The code and a detailed application for these use cases can be found in the Jupyter notebooks `NSRP_test.ipynb` and `STNSRP_test.ipynb` at the GitHub repo (Diez-Sierra et al., 2021a). The objectives and main steps are briefly described here, but we recommend the reader to execute the interactive code

at the Jupyter notebooks while reading this section for a more complete and easier understanding of the applications presented. A small executable is provided to run the examples without previously having to install the library.

    The file `NSRP_test.ipynb` contains a single-site rainfall simulation at hourly and daily scale (although only the latter is presented here) and a dissagregation from daily to hourly rainfall (rainfall downscaling). The file `STNSRP_test.ipynb` contains a multi-site rainfall simulation (commented below) and a multi-site rainfall downscaling example (not included be-

low).

### 4.1   Single-site synthetic rainfall simulation

**Objective**. The library is used here to calibrate the model to reproduce the rainfall characteristics at a specific rainfall station. Once the model is calibrated, it can be used to generate synthetic time series of precipitation showing the same statistical properties as the observations. Such a model would be useful to explore alternative rainfall realizations at the location of

interest, that is, to explore plausible time series of rainfall that may not have been observed due to the limited duration of the observation period. A rainfall station in northern Spain has been selected.

    **Use case configuration**. A monthly analysis is carried out, which means that we assume that the model parameters can be considered homogeneous for any month of the year, but they change from month to month. Hyperparameters for the calibration and for the simulation are reported in the files `Input_Cal_PD.yml` and `Input_Sim_PD.yml`, respectively, at the GitHub

repo. In this case, only one storm system is considered. The average rainfall is given a hundred times more importance (relative weight in the weighted Euclidean distance) in the calibration process that any other statistics. All of the possible statistics included in the NEOPRENE library are used for calibration, to obtain a model that reproduces as well as possible the statistical behavior of the observations. Supra-daily temporal aggregations are selected for some of the statistics in order to simulate longer aggregations which are necessary for hydrological applications. Eighty years of synthetic rainfall data are simulated at

daily and hourly temporal aggregations, although here we focus on the former.

    **Main results** Figure 3 shows the validation of the first use case. The observed, fitted and simulated values for the selected statistics are compared to evaluate the performance of the model. Observed and simulated statistics very close to each other, except for the variance ($\sigma_{nd}$) and skewness ($\bar{\mu}_{3_{1d}}$) for August. For this month, the fitted and observed statistics differ indicating

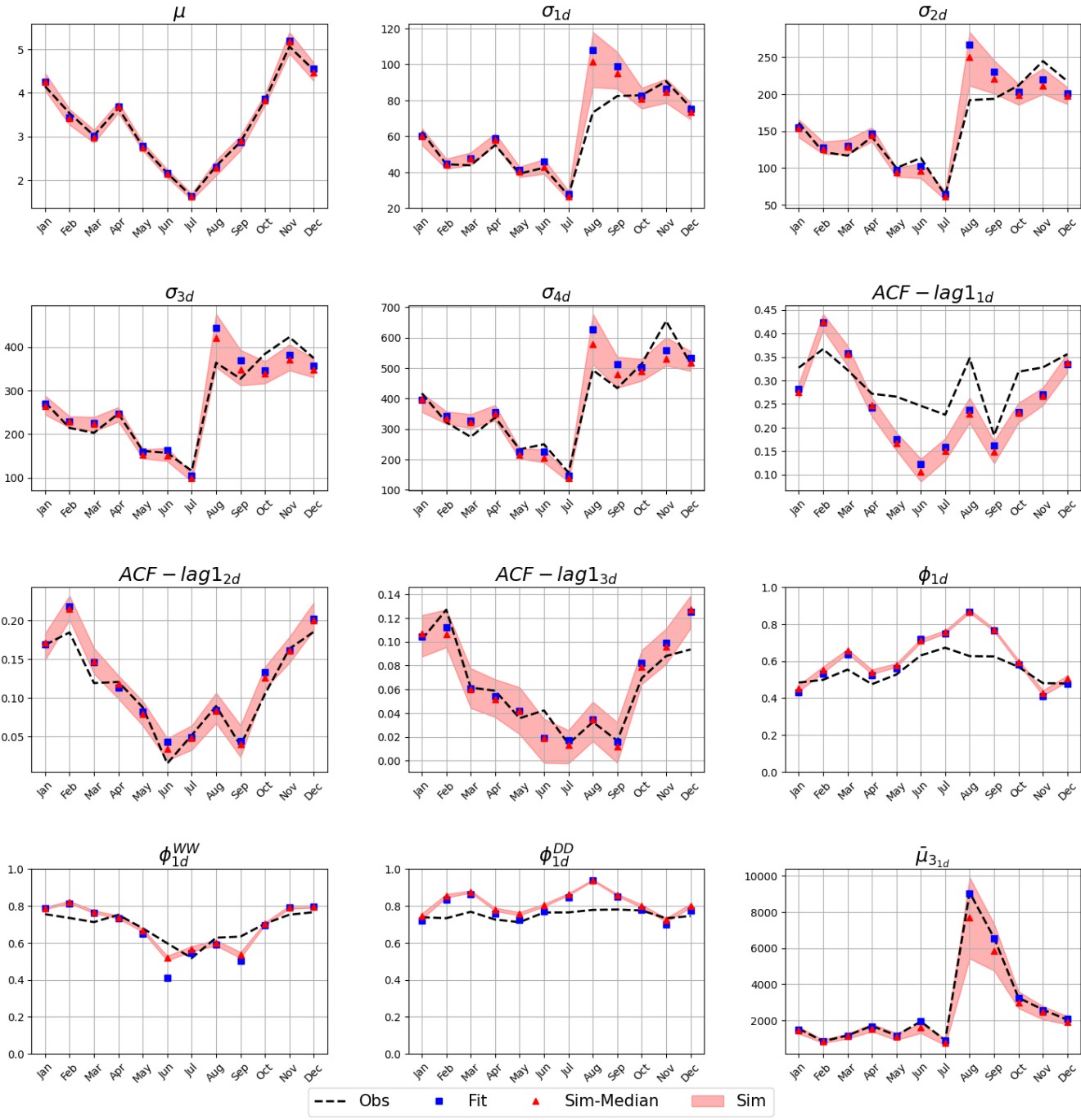

**Figure 3.** Validation plot comparing the observations (dashed line), the fitted (blue squares) and the simulated (red triangles and red shading) statistics, where $\mu$ refers to rainfall average, $\sigma$ to the variance, ACF-lag1 to the autocorrelation of lag one, $\Phi$ to the probability of dry period, $\Phi^{WW}$ to the probability of having two consecutive wet periods, $\Phi^{DD}$ to the probability of having to consecutive dry periods and $(\bar{\mu}_3)$ to the skewness. The subscripts (1d, 2d, etc.) represent the level of aggregation (in days) at which the statistic was computed. The shading shows the range between the 5% and the 95% percentiles. This figure is generated with the `Analysis` python class.

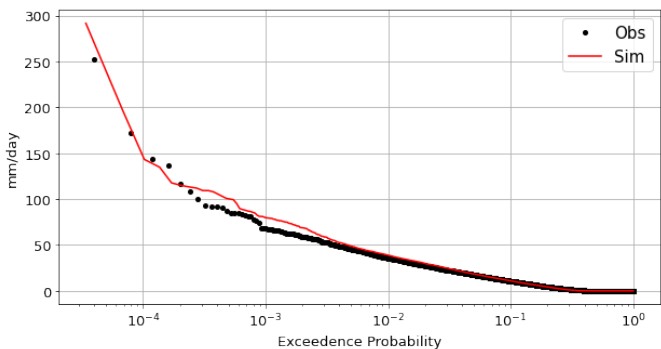

**Figure 4.** Exceedance probability of daily rainfall values for use case #1. Exceedance probability of observed (black dots) and simulated (red line) rainfall values are shown. This figure is generated with the `Analysis` python class implemented for the NEOPRENE library.

that the model parameters fitted are not able to reproduce the observed statistics. In these cases, we recommend to increase the number of calibration iterations or to extend the range of acceptable values for the parameters. If after this modification results do not improve, further analysis should be carried out to test if the model is not able to capture the rainfall properties. As mentioned before, very complex locations, where more than two physical processes are responsible for rainfall, may not be correctly captured by the underlying mathematical model.

The calibrated model can then be used to explore different properties of the rainfall process. For instance, Figure 4 shows an exceedence probability plot, comparing the observed and simulated time series. For high exceedance probability values, this plots can be used as another validation tool. For the local exceedance probability values, however, it can be seen that the simulated values provide a much finer description of the process, so that synthetic generation can be used to better explore the space of extremes, that is, to explore plausible extremes, never observed, but likely to happen.

### 4.2 Rainfall dissagregation

**Objective**. To dissagregate a time series of daily precipitation, producing the most likely hourly time series to have generated the observed daily one, that is, to produce an hourly time series such that when aggregated produces a daily time series as similar as possible to the observed record. Rainfall dissagregation may be an important procedure in forensics analysis of storms, where having a plausible hourly distribution of rainfall may help understand the observed impact of an event for which only an aggregated observation was collected.

**Use case configuration**. Rainfall dissagregation requires to first generate a synthetic time series of rainfall that reproduces the statistics observed —what we did in the first use case—. In this case, the simulation must be at the hourly scale, even when the objective is to reproduce the observed daily statistics.

**Main results** Figure 4 shows an example of the disaggregation results for February 2000. Observed and simulated lines are equal, meaning that the model is able to perfectly reproduce the daily observed precipitation. Black lines shows the hourly disaggretation created with the model.

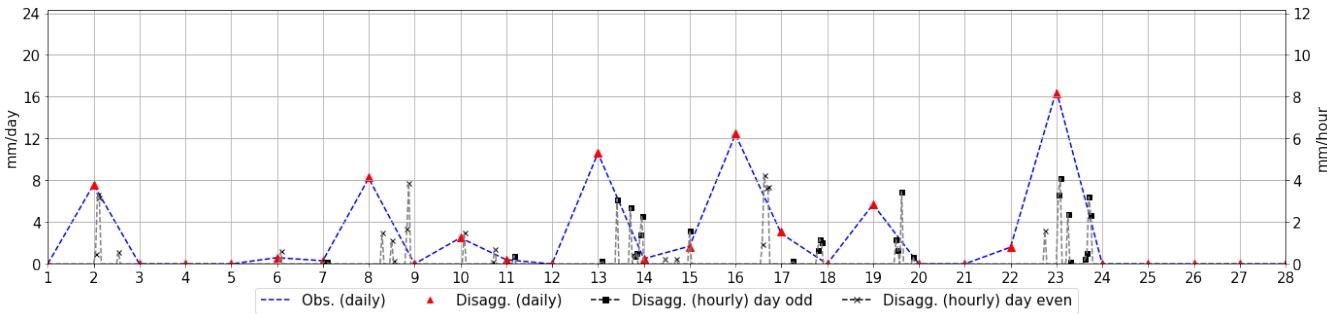

**Figure 5.** Disaggretation plot for the rainfall of the month of February 2000. Blue (observation) and red (simulated) lines correspond to the observed and simulated series at daily scale, respectively. Back lines shows the hourly disaggretation simulated with the model for odd (square) and even (asterisk) days. This figure is generated with the `Analysis` python class.

### 4.3 Multi-site synthetic rainfall simulation

**Objective**. The library is used here to calibrate the model to reproduce the average rainfall characteristics for a collection of rainfall time series. Once the model is calibrated, it can be used to generate multi-site synthetic time series of precipitation that follow the same statistical properties of the input time series. Note that simulated series reproduce the average rainfall statistics calculated with the entire collection of observed rainfall time series except for the mean which fits to each location. Several gages from a basin located in northern Spain were selected.

Use case configuration. A seasonal analysis is carried out, which means that we assume that the model parameters can be considered homogeneous for any given season (winter, spring, summer and fall), but they change from season to season. Hyperparameters for the calibration and for the simulation are reported in the files `Input_Cal_SPD.yml` and `Input_Sim_SPD.yml`, respectively, at the GitHub repo. Similarly to the first use case, only one storm system is considered. The cross-correlation is given ten times more importance (relative weight in the weighted Euclidean distance) in the calibration process that any other statistic. One hundred years of synthetic rainfall data are simulated for each one of the gages at both daily and hourly temporal aggregations.

Main results Figure 6 shows an exceedance probability plot, comparing the exceedance probability of observed and simulated rainfall values aggregated for the collection of rainfall time series. The results for the observed and simulated time series are very similar, proving the capabilities of NEOPRENE to reproduce maximum observed aggregated rainfall events and, thus that it is an useful tool for flood analysis.

Finally, Figure 7 shows a comparison of the observed, fitted and simulated results for the cross-correlation. The observed and simulated cross-correlations are empirically computed from the time series, while the calibrated cross-correlation is computed using analytic expressions. While some exact analytic expression exists (for the mean and the variance, for instance), they do not exists for all the statistics. Indeed, some statistics require some approximations and series expansions that induce the behavior shown in the figure: the calibration values approximate quite well the observed ones, but the simulated ones presents

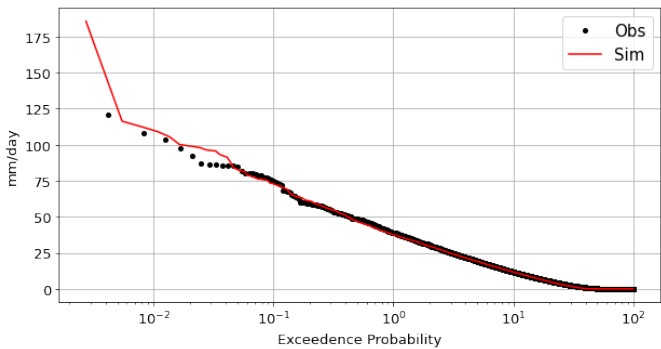

**Figure 6.** Exceedance probability of daily rainfall values for use case #3. Exceedance probability of observed (black dots) and simulated (red line) rainfall values are shown. Note that the figure shows the exceedance probability averaged over all the rainfall series. This figure is generated with the `Analysis` python class.

a small bias. For most practical purposes, the simulated series reproduce adequately the observed cross-correlation, but the fit should be analyzed to verify that differences are kept below acceptable thresholds.

## 5   Discussion

NEOPRENE constitutes a user-friendly tool for spatio-temporal synthetic rainfall generation based on the Neyman-Scott process. Compared with other statistical approaches for synthetic rainfall generation such as Probability Distribution Models or Markov Chain Models (Wilks, 1998), Point processes, like the Neyman-Scott, are more efficient capturing the temporal and spatial dependence of rainfall and reproducing different rainfall regimes, particularly the extreme events' one. However, it has the disadvantage of requiring more computational resources and some knowledge of its internals. Compared with Artificial Neural Networks (Welten et al., 2022), Point processes may be less flexible in terms of incorporating non-linear relationships or external information. However, ANN are still not widely used as rainfall synthetic generators, being more commonly used for rainfall-runoff prediction.

Particularity, NEOPRENE has been validated to reproduce hourly and daily return periods in hundreds of gauges in Spain. Furthermore, its implementation removes the main hindrance to the practical application of the model, which is related to the complexity of model parameter estimation (Onof et al., 2000). It is important to point out that the properties of the spatial model present some limitations. First, because it requires some supervision in order to find a suitable model adjustment, and second, because it is based on a number of assumptions, such as on homogeneity, which makes it not appropriated for locations where statistics other than the mean are not homogeneous.

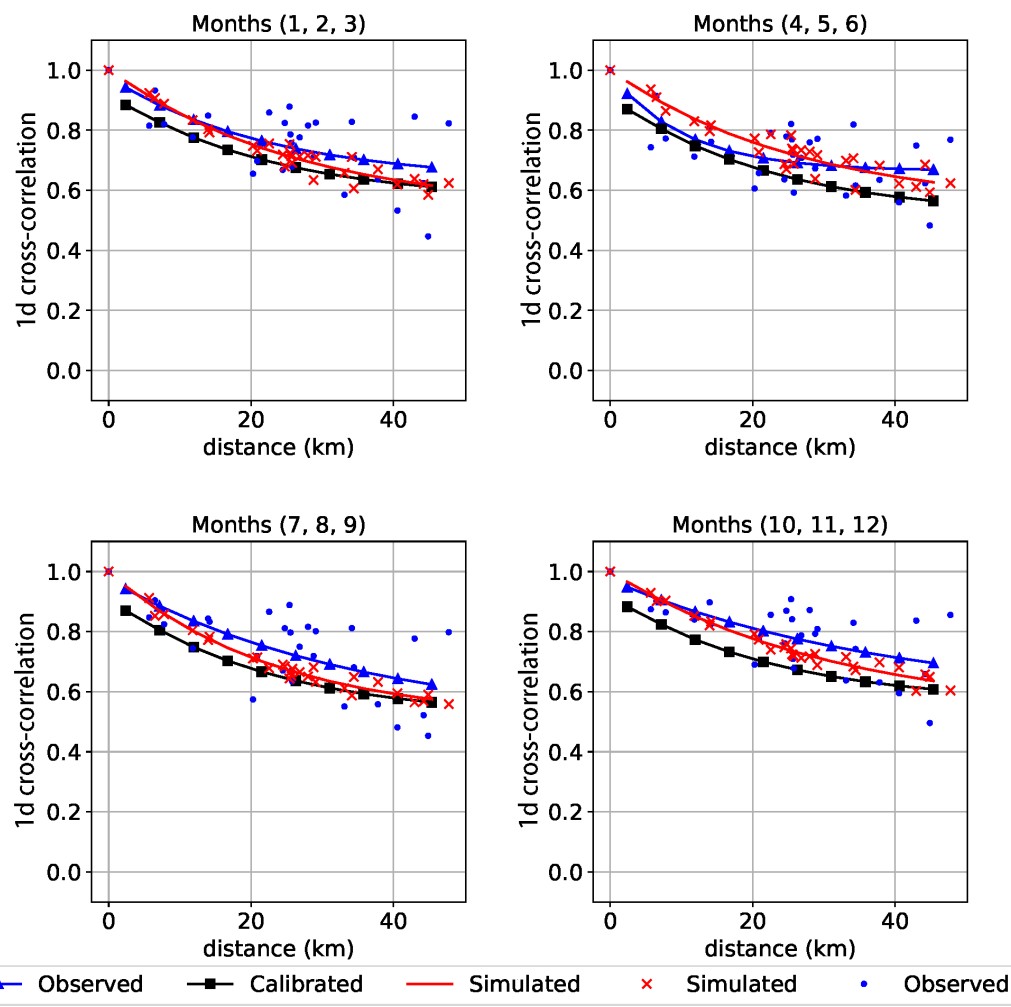

**Figure 7.** Validation plot comparing the observations (blue), fitted (black) and the simulated (red) cross-correlation. This figure is generated with the `Analysis` python class implemented for the NEOPRENE library.

## 6 Future challenges

Although the current implementation of NEOPRENE already provides useful tools for research and hydraulic engineering practice, we plan on improving the functionality of the library. The main points to improve in the near future are:

- **Parameter smoothing across months**: Currently, the time decomposition makes all the fits independent of one another. We are planning to implement a hierarchical scheme that properly weighs the influence of other close-in-time decomposition units.

- **Storm radius parameter**: NEOPRENE does not consider yet a reduction of far away correlations due to the limited size of the storm. This is only a limitation when huge domains are involved.

- **Sub-hourly implementation**: The hourly time scale is detailed enough for most applications. However, we plan on exploring the generation of sub-hourly time series.

- **Virtual gauges**: We will implement an interpolation technique based on the underlying mathematical model, allowing to incorporate the fitted structure of the rainfall field into the interpolation procedure.

- **Raster input / output**: Currently, NEOPRENE only works with rainfall gages. However, a next step would be to allow it to ingest rainfall raster data (satellite or radar ) and to also produce them.

We expect that the GPLv3 license of the library and the fact that it is readily on GitHub will attract external collaborators that will help to improve the functionality of the library even further.

## 7 Conclusions

We have presented NEOPRENE, an open-source Python library for generating synthetic rainfall fields using the Neyman-Scott process. The library allows to generate rainfall at different temporal scales of aggregation to match rainfall observations. The library is available at GitHub (Diez-Sierra et al., 2021a) and Zenodo (Diez-Sierra et al., 2021b), under a free license (GPLv3). Therefore, it can be freely used for research and commercial purposes.

NEOPRENE can be used for multiple purposes such as water resources assessment, extreme rainfall analysis or rainfall disaggregation, among others. NEOPRENE is designed to reproduce second order moments and allows to simulate two storm systems simultaneously to capture different rainfall generation processes (i.e. frontal and convective precipitation).

Jupyter notebooks provide an easy entry point to the library, presenting its most important functionality, and converting it in an accessible tool for many sector professionals (hydrologists, hydraulic engineers and climate practitioners). Special attention has been placed in demonstrating the ability of NEOPRENE to reproduce observed extreme events, because it makes NEOPRENE specially useful in Engineering practice (e.g. return period estimation for flood analysis).

*Code and data availability.* The NEOPRENE Python library code is available at GitHub (https://github.com/IHCantabria/NEOPRENE) and Zenodo (https://doi.org/10.5281/zenodo.6349377) under the GNU General Public License version 3.0. Data used in this work is also available from the same sources.

## Appendix A:  Some statistical properties of the model

In model fitting, it is usually necessary to use equations for aggregated properties, because rainfall data are usually sampled over discrete time intervals. Let $Y_{ij}^h(\boldsymbol{x})$ be the aggregated time series of rainfall due to type $i$ storms at point $\boldsymbol{x} = (x, y) \in R^2$ in the $j - th$ time interval of width $h$, and let $Y_j^h(\boldsymbol{x})$ be the total rainfall in the $j - th$ interval due to the superposition of the $n$

storm types (Cowpertwait et al., 2013). Then,

$$Y_j^h(\boldsymbol{x}) = \sum_{i=1}^{n} \int_{(j-1)/h}^{jh} Y_i(\boldsymbol{x}, t) dt \tag{A1}$$

where $Y_i(\boldsymbol{x}, t)$ is the rainfall intensity at point $\boldsymbol{x}$ and time $t$ due to type $i$ storms $(i = 1, ..., n)$. Since the superposed processes are independent, statistical properties of the aggregated time series follow just by summing the various properties given below.

**Mean (Cowpertwait, 1991, Eq. 12):**

$$\mu(h) = \sum_{i=1}^{n} E\{Y_{ij}^h(\boldsymbol{x})\} = h \sum_{i=1}^{n} \frac{\lambda_i \nu_i}{\chi_i \eta_i} \tag{A2}$$

in the STNSRP model, $\chi_i$ must me change by $\chi_i^- 1$

**Covariance (Cowpertwait, 1991, Eq. 14):**

$$\gamma(\boldsymbol{x}, \boldsymbol{x}, l, h) = \sum_{i=1}^{n} Cov\{Y_{ij}^h, Y_{i,j+l}^h\} = \frac{\lambda_i(\nu_i^2 - 1)[\beta_i^3 A_i(h, l) - \eta_i^3 B_i(h, l)]}{\beta_i \xi^2 \eta_i^3 (\beta_i^2 - \eta_i^2)} - \frac{4\lambda_i \chi_i A_i(h, l)}{\xi_i^2 \eta_i^3} \tag{A3}$$

when $l = 0$,

$A_i(h, l) = A_i(h) = \eta_i h - 1 + e^{-\eta_i h}$       (A4)

$B_i(h, l) = B_i(h) = \beta_i h - 1 + e^{-\beta_i h}$       (A5)

when $l > 0$,

$A_i(h, l) = 0.5(1 - e^{-\eta_i h})^2 \exp^{-\eta_i h(l-1)}$       (A6)

$B_i(h, l) = 0.5(1 - e^{-\beta_i h})^2 \exp^{-\beta_i h(l-1)}$       (A7)

**Probability of not rain in an arbitrary time of length $h$ (Cowpertwait et al., 1996a, Eq. 6):**

The probability that an arbitrary time interval $[(j-1)h, jh]$ is dry at a point is obtained by multiplying the probabilities of the independent processes and is given by the following:

$$\phi(h) = \sum_{i=1}^{n} Pr\{Y_{ij}^{(h)}(\boldsymbol{x}) = 0\} = exp(-\lambda_i h + \lambda_i \beta_i^{-1}(\nu_i - 1)^{-1}\{1 - exp[1 - \nu_i + (\nu_i - 1)e^{-\beta_i h}]\} - \lambda_i \int_0^\infty [1 - p_h(t)]dt) \quad (A8)$$

We use the approximation shown in (Cowpertwait, 1991, Eq. 17) to avoid having to solve the integral:

$$\int_0^\infty [1 - p_h(t)]dt = \beta_i^{-1}[\gamma + ln(\alpha \nu_i - 1)] \quad (A9)$$

where $\gamma = 0.5771$ and $\alpha_i = \eta_i/(\eta_i - \beta_i) - e^{-\beta_i h}$

**Transition probabilities**:

The transition probabilities, $pr\{Y_{i,j+1}^{(h)}(x) > 0 | Y_{ij}^{(h)}(x) > 0\}$ and $pr\{Y_{i,j+1}^{(h)}(x) = 0 | Y_i^{(h)}(x) = 0\}$, denoted as $\phi_{WW}(h)$ y $\phi_{DD}(h)$, respectively, can be expressed in terms of $\phi(h)$ following (Cowpertwait et al., 1996a, Eq. 7, 8 and 9):

$$\phi_{DD}(h) = \phi(2h)/\phi(h) \quad (A10)$$

$$\phi(h) = \phi_{DD}(h) + \{1 - \phi_{WW}(h)\}\{(1 - \phi(h))\} \quad (A11)$$

$$\phi_{WW}(h) = \{1 - 2\phi(h) + \phi(2h)\}\{1 - \phi(h)\} \quad (A12)$$

**The third moment function (Cowpertwait, 1998, Eq. 10):**

$$\xi_h = E\{Y_j^{(h)}(\boldsymbol{x}) - \mu(h)\}^3 = \sum_{i=1}^{n} [6\lambda_i \nu_i E(X^3)(\eta_i h - 2 + \eta_i h e^{-\eta_i h} + 2e^{-\eta_i h})/\eta_i^4 \quad (A13)$$

$$+ 3\lambda_i \chi_i E(X^2)\nu_i^2 f(\eta_i, \beta_i, h)/\{2\eta_i^4 \beta_i(\beta_i^2 - \eta_i^2)^2\}$$

$$+ \lambda_i \chi_i^3 \nu_i^3 g(\eta_i, \beta_i, h)/\{e\eta_i^4 \beta_i(e\eta_i^2 - \beta_i^2)(\eta_i - \beta_i)(2\beta_i + \eta_i)(\beta_i + 2\eta_i)\}]$$

In the STNSRP model, C follows a Poisson random variable, so that $E\{C(C-1)\} = \nu^2$ and $E\{C(C-1)(C-2)\} = \nu^3$. If it follows a geometric one then $E\{C(C-1)\} = 2\nu^2(\nu - 1)$ and $E\{C(C-1)(C-2)\} = 6\nu^2(\nu - 1)^2$

For exponential cell intensities, $E(X_{ijk})$ and $E(X_{ijk}^k)$ are replaced by $2\chi_i^2$ and $6\chi_i^3$, respectively. $f(\eta_i, \beta_i, h)$ and $g(\eta_i, \beta_i, h)$ are derived below:

$$f(\eta_i, \beta_i, h) = -2\eta^3\beta^2 \exp(-\eta h) - 2\eta^3\eta^2 \exp(-\beta h) + \eta^2\beta^3 \exp(-2\eta h) + 2\eta^4\beta \exp(-\eta h) + 2\eta^4\beta \exp(-\beta h) \quad (A14)$$

$$+ 2\eta^3\beta^2 \exp(-(\eta + \beta)h) - 2\eta^4\beta \exp(-(\eta + \beta)h) - 8\eta^3\beta^3 h + 11\eta^2\beta^3 - 2\eta^4\beta + 2\eta^3\eta^2 + 4\eta\beta^5 h$$

$$+ 4\eta^5\beta h - 7\beta^5 - 4\eta^5 + 8\beta^5 \exp(-\eta h) - \beta^5 \exp(-2\eta h) - 2h\eta^3\beta^3 \exp(-\eta h) - 12\eta^2\beta^3 \exp(-\eta h)$$

$$+ 2h\eta\beta^5 \exp(-\eta h) + 4\eta^5 \exp(-\beta h)$$

$$g(\eta_i, \beta_i, h) = 12\eta^5\beta\exp(-\beta h) + 9\eta^4\beta^2 + 12\eta\beta^5\exp(-\eta h) + 9\eta^2\beta^4 + 12\eta^3\beta^3\exp(-(\eta+\beta)h) - \eta^2\beta^4\exp(-2\eta h) \quad \text{(A15)}$$

$$- 12\eta^3\beta^3\exp(-\beta h) - 9\eta^5\beta - 9\eta\beta^5 - 3\eta\beta^5\exp(-2\eta h) - \eta^4\beta^2\exp(-2\eta h) - 12\eta^3\beta^3\exp(-\eta h)$$

$$+ 6\eta^5\beta^2 h - 10\beta^4\eta^3 h + 6\beta^5\eta^2 h - 10\beta^3\eta^4 h + 4\beta^6\eta h - 8\beta^2\eta^4\exp(-\beta h) + 4\beta\eta^6 h + 12\beta^3\eta^3$$

$$- 8\beta^4\eta^2\exp(-\eta h) - 6\eta^6 - 6\beta^6 - 2\eta^6\exp(-2\beta h) - 2\beta^6\exp(-2\eta h) + 8\eta^6\exp(-\beta h)$$

$$+ 8\beta^6\exp(-\eta h) - 3\beta\eta^5\exp(-2\eta h)$$

For each storm, the number of cells $\nu$ that overlap a point in $R^2$ is a Poisson random variable with mean (Cowpertwait et al., 2002, Eq. 3):

$$\nu_C = \nu\phi_c^2/(2\pi) \quad \text{(A16)}$$

where $\nu_C$ denotes the two-dimensional Poisson process (cells per $km^2$)

The probability that a cell overlaps a point $x$ given that it overlapped a point $y$ a distance $d$ from $x$ (Cowpertwait et al., 2002, Eq. 9)

$$P(\phi, d) \approx \frac{1}{30}\sum_{i=1}^{4}\{2f(\frac{2\pi i}{20}) + 4f(\frac{2\pi i + \pi}{20})\} - \frac{1}{30}f(0) \quad \text{(A17)}$$

where

$$f(y) = (\frac{\phi d}{2\cos y} + 1)exp(\frac{-\phi d}{2\cos y}), 0 \leq y < \pi/2, f(\pi/2) \quad \text{(A18)}$$

**Cross-correlation (Cowpertwait et al., 2002, Eq. 6):**

$$\gamma(x, y, l, h) = \sum_{i=1}^{n}Cov\{Y_{ij}^h(x), Y_{i,j+l}^h(y)\} = \sum_{i=1}^{n}[\gamma_i(x, x, l, h) - 2\lambda_i\{1 - P(\phi_i, d)\}\nu_{Ci}E(X^2)A_i(h, l)/\eta_i^3] \quad \text{(A19)}$$

*Author contributions.* Javier Diez-Sierra contributed to the Conceptualization, Investigation, Formal Analysis, Software Development, Validation and Writing. Salvador Navas contributed to the Software Development and Validation. Manuel del Jesus contributed to the Conceptualization, Software Development, Writing, Supervision and Funding Acquisition

*Competing interests.* The authors declare to have no competing interests.

*Acknowledgements.* Salvador Navas acknowledges the financial support from the Government of Cantabria through the Fénix Programme. Manuel del Jesus acknowledges the funding provided by Grant RTI2018-096449-B-I00 funded by MCIN/AEI/10.13039/501100011033 and by "ERDF A way of making Europe".

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
