# Peer review of "NEOPRENE v1.0.1: A Python library for generating spatial rainfall based on the Neyman-Scott process"

_EGUsphere, 2022_

## Author Response (AR1)

**NEOPRENE v1.0.1: A Python library for generating spatial rainfall based on the Neyman-Scott process**
**Response to the reviewers**

Javier Diez-Sierra, Salvador Navas and Manuel del Jesus

May 9, 2023

Dear Editors and Reviewers,

In this letter we provide answers to the comments posted in the interactive discussion of the paper referenced in the title. We believe that our answers and modifications of the paper solved the issues that were raised, but we are open to discuss those points that may not still be completely clear, and to further modify those that need to.

We appreciate your comments and provide our answers in the following lines.

**1   Executive Editor of GMD**

1. **Dear authors,**
   **please make the effort to add the citations to the Zenodo and GitHub repositories in the Code availability section again. Best regards, Astrid Kerkweg (Exectuive Editor of GMD)**

   We have included the citations as you suggested. Now both urls appear in the code availability section.

**2   Reviewer 1**

Dear Reviewer,

We would like to thank you for your comments. In the following lines we provide answers to them.

1. **Line 16: There are several other studies that used the weather generator for disaster risk analysis such as:**

   We appreciate the references that you provide, and we consider them interesting so we have included them in the revised manuscript.

2. **Line 19: Poisson clustered**

   We have not fully understood if your comment was cut or if it was a spelling correction. We have checked and we believe that the spelling was correct, but please, do not hesitate to let us know if the issue was other one.

3. **Line 28: There are also many papers regarding Bartlett-Lewis type model, but references regarding Neyman-Scott model are provided. Please consider citing following articles:**

   You are right. We have not included Bartlett-Lewis references. As the four references that you suggest are relevant and interesting, we have included them in the revised manuscript. Thank you for the suggestion.

4. **Line 35: There are some other tools too such as:**

   As we mentioned in the paragraph, we were only aware of the tools that we have cited. We have included a comment on Let-It-Rain in the revised manuscript.

5. **Line 113-115: PSO is not actually the best solution for the calibration. For the next version of this application, please try the simplex algorithm as follow:**

   We are aware that multidimensional calibration or optimization are arduous tasks that lead to multiple problems, like multimodality that you mention. In general, PSO has served us well when dealing with uncertainty analysis. We have found that it is comparable to other optimization methods (e.g. Simulated Annealing) but allowed some extra flexibility in the information exchange that in some cases was useful.

   We will study the reference that you have shared with us and will consider to implement it for a future version of NEOPRENE.

6. **Figure 3. I suggest authors to draw the inter-annual variability of the monthly statistics using the shades in the existing plot (e.g. repeat the simulation for 100 times and plot the mean, upper and lower boundary (e.g. interquartile range) of the statistics of the 100 simulation, and then compare this shades to those corresponding to observed rainfall). This is because the rainfall generator's primary purpose is risk analysis through Monte-Carlo simulation based on multiple simulations. You may also notice that the shade of the simulation is a lot thinner than that of observation because Poisson cluster rainfall model is designed to reproduce fine-scale rainfall statistics only (e.g. less than a couple of days). This comment is also concerned with Section 5. Future Challenges.**

Based on your comment we have created two different plots (attached to the comment in a Zip file). In the first one ($\text{Figure3}_{\text{Range}}$), we simply analyze the variability that we obtain from 100 different simulations when analyzing the statistics. There you can see that the shaded region is quite thin, that is, that the long-term statistics are consistently reproduced by the model. As it can be observed in the original Figure 3, the summer months present more difficulties, due to the amount of zeros in the time series, but the aggregation of multiple years makes the statistics more or less stable.

In the second plot ($\text{Figure3}_{\text{Interannual}}$) we have reproduced the analysis that you suggest, that is, we have analyzed the inter-annual variability of the different statistics of the observed time series (gray shading) and of the simulations (red shading). The simulations consider 100 realizations, so the shadings are not completely equivalent, but nonetheless they can be compared.

As you can see, in general terms the model captures an inter-annual variation range which is quite similar to the observed one. In the summer months we have larger problems, and more noise, but this is due to the fact that we are computing statistics out of a month observation, that is, only 30 data points, which may turn the computation of some statistics quite unstable.

Indeed, in this figure we are representing 12 statistics computed from 30 data points, what may be introducing some noise in the representation

and some of the features may not be fully representative.

In any case, the original figure 3 was intended as a validation plot of the quality of the model. We believe that the plot comparing the model variability to the observations may provide a fuller image of this comparison. The inter-annual plot is more complete in some respects, but may be more complicated to explain. We are open to include the inter-annual version, together with an analysis of why capturing the inter-annual variability range may be important for risk analysis, but we would like to request your opinion after seeing the results to be sure that we understood the point of your comment.

**3 Reviewer 2**

Dear Reviewer,

We would like to thank you for your kind words about our work and for your comments, that will improve the overall quality of the paper. In the following lines we provide answers to them.

1. **In my opinion, it would be helpful to provide more specific feedback on the strengths and limitations of the Neumann-Scott model especially in comparison with other statistical models of precipitation that could be used as alternative to the present approach.**

   To address this point, we have included a new section, "Discussion", in the manuscript to provide more specific feedback on the strengths and limitations of the model in comparison with other methods for synthetic rainfall generation. The section reads as follows:

   *NEOPRENE constitutes a user-friendly tool for spatio-temporal synthetic rainfall generation based on the Neyman-Scott process, which is a point process. Compared with other statistical approaches for synthetic rainfall generation such as Probability Distribution Models or Markov Chain Models (Wilks, D. S. et al. 1998), Point processes are more efficient capturing the temporal and spatial dependence of rainfall and reproducing different rainfall regimes, particularly the one of extreme events. However, it has the disadvantage of requiring more computational resources and some knowledge of its internals. Comparing with*

*Artificial Neural Networks (ANN, Welten, S. 2022), Point processes may be less flexible in terms of incorporating non-linear relationships or external information. However, ANN are still not widely used as rainfall synthetic generators, being more commonly used for rainfall-runoff prediction.*

*Particularity, NEOPRENE has been validated to reproduce hourly and daily return periods in hundreds of gauges in Spain. Furthermore, its implementation removes the main hindrance to the practical application of the model, which is related to the complexity of model parameter estimation (Onof et al. 2000). It is important to point out that the properties of the spatial model present some limitations. First because it requires some supervision in order to find a suitable model adjustment, and second because it is based on a number of assumptions, such as on spatial stationarity, which makes it not appropriated for locations where statistics other than the mean are not stationary.*

+ Onof C, Chandler RE, Kakou A, Northrop P, Wheater HS, Isham V (2000) Rainfall modelling using Poisson-cluster processes: a review of developments. Stoch Env Res Risk Assess 14(6):384–411

+ Wilks, D. S. (1998). "Multisite generalization of a daily stochastic precipitation generation model." Journal of Hydrology, 210(1-4), 178-191

+ Sascha Welten, Adrian Holt, Julian Hofmann, Lennart Schelter, Elena-Maria Klopries, Thomas Wintgens, Stefan Decker (2022). Synthetic rainfall data generator development through decentralised model training, Journal of Hydrology, Volume 612, Part C, 2022, 128210, ISSN 0022-1694, `https://doi.org/10.1016/j.jhydrol.2022.128210`.

2. **Line 75 - 80: It would be helpful to provide the formulas in an appendix or supplemental material, as some readers may want to reference them.**

The statistical description of the model is included in the GitHub repository (we make a reference to this material in line 65). We left these formulas in the GitHub repo to save space in the paper and make it a modifiable part of our work, that could be changed with future updates.

However, if you believe that having these formulas as an appendix could be useful, we will include them as an appendix in the revised

manuscript.

3. **Line 95: The concept of cell radius and cell intensity may not be clear to all readers, so it would be helpful to provide more explanation or context.**

We have modified and updated the paragraph in line 72 to include a better explanation of the geometry of the rain cells that form each storm. We believe that this explanation explains better both concepts. The paragraph now reads as follows:

*The interarrival time between the origins of storm of type i follows and exponential distribution with paramater $Lambda_i$. Each storm is composed of several rainfall cells, which form a marked point process. Each rainfall cell is assumed to be circular and to contain a random amount of water, that produces rainfall at a random rate (cell intensity) during the lifetime of the rainfall cell (cell duration). The superposition of all these cells and their combined intensities produces the total rainfall intensity of the model. The marked point process generated by the rainfall cells is characterized by:*

4. **Line 105: Please provide the URL of the repository where readers can download the Neumann-Scott model implementation.**

We have not included URLs in the main text of the manuscript since we believe that they tend to complicate the reading of the manuscript. That is why we have opted to include them as a citation or a reference to the webpage.

However, following also another review comment, we will include the complete URL in the *code availability* section so that the URL is easy to find and the reader does not need to look for it in the references.

5. **Line 185: Please provide more detail on how the seasonality of rainfall is reproduced in the model.**

We have included more detail on how seasonality is reproduced in the model in section 2.3. Below we copy the text that we have added:

*Seasonality is included in the model accounting for different sets of model parameters and observed statistics. For instance, a single set of parameters will be used to calibrate and generate from a model without*

*seasonality, where years are assumed to be a stationary period. However, in regions where two well differentiated periods may be observed -a wet and a dry season, for instance-, two different sets of parameters will be used: one for the wet season and another one for the wet season. That is, seasonality is accounted for by decomposing the complete time series into subseries that only contain the information related to the desired season or time period.*

6. **Line 215: It would be helpful to provide a clear definition of what is meant by a "rain event" within a storm.**

   We have changed *rain event* by *rain cell* which is the correct term and that we believe does not generate confusion.

7. **Line 235: The phrase about the analysis module not providing specific statistical tests to verify the goodness of fit is unclear. Please provide more explanation or context.**

   We have modified the paragraph to clarify the idea that we wanted to convey. The new text reads as follows:

   *To evaluate the quality of the model fit, the user should determine which statistical test may be appropriate in every case. A Kolmogorov-Smirnov test could be suitable to test if the rainfall generated and the observed one come from the same distribution, while a t-test could serve to analyze if the difference in mean precipitation is significant. The number of possible hypothesis tests that may be carried out over model results is huge, since different applications may require different quality evaluation. Therefore, we do not provide any goodness-of-fit routine or specific hypothesis testing within the Analysis module. However, we do provide functions to help to inspect the quality of the fits.*